# Newborn care knowledge and practices among care givers of newborns and young infants attending a regional referral hospital in Southwestern Uganda

Dorah Nampijja[1,2]*, Stella Kyoyagala[1,2], Elizabeth Najjingo[2], Josephine N. Najjuma[1], Onesmus Byamukama[2], Lydia Kyasimire[1], Jerome Kabakyenga[1], Elias Kumbakumba[1]

1 Mbarara University of Science and Technology, Mbarara, Uganda, 2 Mbarara Regional Referral Hospital, Mbarara, Uganda

* drdorah@yahoo.com

## Abstract

A child born in developing countries has a 10 times higher mortality risk compared to one born in developed countries. Uganda still struggles with a high neonatal mortality rate at 27/1000 live births. Majority of these death occur in the community when children are under the sole care of their parents and guardian. Lack of knowledge in new born care, inappropriate new born care practices are some of the contributors to neonatal mortality in Uganda. Little is known about parent/caregivers' knowledge, practices and what influences these practices while caring for the newborns. We systematically studied and documented newborn care knowledge, practices and associated factors among parents and care givers. To assess new born care knowledge, practices and associated factors among parents and care givers attending MRRH. We carried out a quantitative cross section methods study among caregivers of children from birth to six weeks of life attending a regional referral hospital in south western Uganda. Using pretested structured questionnaires, data was collected about care givers' new born care knowledge, practices and the associated factors. Data analysis was done using Stata version 17.0. We interviewed 370 caregivers, majority of whom were the biological mothers at 86%. Mean age was 26 years, 14% were unemployed and 74% had monthly earning below the poverty line. Mothers had a high antenatal care attendance of 97.6% and 96.2% of the deliveries were at a health facility Care givers had variant knowledge of essential newborn care with associated incorrect practices. Majority (84.6%) of the respondents reported obliviousness to putting anything in the babies' eyes at birth, however, breastmilk, water and saliva were reportedly put in the babies' eyes at birth by some caregivers. Hand washing was not practiced at all in 16.2% of the caregivers before handling the newborn. About 7.4% of the new borns received a bath within 24 hours of delivery and 19% reported use of herbs. Caregivers practiced adequate thermal care 87%. Cord care practices were inappropriate in 36.5%. Only 21% of the respondents reported initiation of breast feeding within 1 hour of birth, Prelacteal feeds were given by 37.6% of the care givers, water being the commonest prelacteal feed followed by cow's milk at 40.4 and 18.4% respectively. Majority of the respondents had below average knowledge about danger signs in the

**Data Availability Statement:** The data that was collected and used for the study will be made available on request from the MUST research Ethics committee of Mbarara University of Science and Technology at uncstresearch@uncst.go.ug and palele@must.ac.ug.

**Funding:** This study received funding from MGH first mile SEED research grant to DN. The funders had no role in study design, data collection and analysis, decision to publish, or preparation of the manuscript.

**Competing interests:** The authors have declared that no competing interests exist.

newborn where 63% and mean score for knowledge about danger signs was 44%. Caretaker's age and relationship with the newborn were found to have a statistically significant associated to knowledge of danger signs in the newborn baby. There are variable incorrect practices in the essential new born care and low knowledge and awareness of danger signs among caregivers of newborn babies. There is high health center deliveries and antenatal care attendance among the respondents could be used as an opportunity to increase caregiver awareness about the inappropriate practices in essential newborn care and the danger signs in a newborn.

## Introduction

Globally, more than 8 million children die before their 5[th] birthday [1], and neonatal deaths accounts for half of all under five mortality [2]. Every year 2.5 million neonates die and majority of these deaths occur in low and middle income countries [3]. A child born in developing countries has a 10 times higher mortality risk compared to one born in developed countries [4,5] Sub Saharan Africa has the highest neonatal mortality rate in the sustainable development goal regions at 28/1000 live birth, and 1.2 million neonatal deaths every year, majorly from preventable causes like hypothermia and poor neonatal practices [3,6]. Uganda's neonatal mortality rate has stagnated at 27/1000 live birth over the past 10 years [7], and large number of new born deaths occur in the communities under the sole care of their parents and care takers [8–10].

WHO recommends essential newborn care including early initiation of breast feeding, keeping babies warm, recognition of neonatal danger signs and cord care, among others, as crucial in new born survival [11,12]. Most parents care for their new born babies with knowledge acquired from friends and family which may be harmful and incorrect [13]. Once a baby is born in the community or discharged from a health facility, his/her care is entirely in the hands of the parents (often the mother) and the care givers. Lack of knowledge in new born care, inappropriate new born care practices like false teeth extraction, harmful cord care, prelacteal feeding by caretakers and low health center deliveries contribute to the high neonatal deaths [10,13,14].

Understanding newborn care knowledge and practices among parents and care givers is crucial to improve survival among newborns. Parent centered newborn care packages have been studied in some countries and have proved beneficial in newborn outcomes [15]. There is no available parent centered programs for newborn care in Uganda and little is known about what parents /caregivers know or do and what influences their actions while they care for their newborn babies at home. We set out to assess new born care knowledge, practices and associated factors among parents and care givers to newborn babies at Mbarara Regional Referral Hospital. This information will help to design feasible and acceptable high impact new born care packages and set a baseline for health education for parents and care givers of new born children.

## Methods

### Study area

We conducted a cross sectional, quantitative hospital based study from November 2022 to February 2023 at a regional referral hospital in South western Uganda with a catchment population of 3 million people with a bed capacity of 494 beds. The hospital also serves as a teaching

hospital for Mbarara University of Science and Technology (MRRH). The hospital offers a number of services including antenatal care, delivery and obstetric care, paediatric care including a newborn and premature care unit. The Paediatric Newborn Unit has a total bed occupancy of 2.5 above capacity and on average 2000 neonates are admitted annually. There are no standardized newborn care packages that are designed for parents and caregivers of newborn infants before or after birth at the hospital. Using a study done on Newborn Care Practice and Associated Factors among Mothers of One-Month-Old Infants in Southwest Ethiopia [16], we included 370 participants. The participants were either caretakers or parents of children at the immunization clinic, maternity ward and newborn unit at MRRH. Ethical approval was obtained from Mbarara University of science and Technology Institutional review board was obtained before commencement of the study (REC No. MUST -2022-600). Written informed consent was obtained from all participants before they could participate in the study. Parents and care givers of very sick babies, whose children had died, or were sick or attending to sick mothers were excluded from the study until they were stable. We used a pre tested questionnaire to collect data on demographics, essential newborn care knowledge and practices from parents and care givers of children from birth to 6 weeks who were attending the neonatal unit and clinic, postnatal ward and the maternal and child health clinic at the referral hospital. Demographic data of the care givers, and information on their knowledge and practices about the essential care of the new born (ECNB), age, education status, income, parity, ANC attendance, family status, birth support partner was collected. New born care knowledge practices including breastfeeding (when to initiate breast feeding, frequency, prelacteal feeds, colostrum feeding), cord care, immunization, management of colic, cultural beliefs, false teeth extraction, child scarification, use of herbs were assessed and presented as percentages and proportions. Recognition of danger signs (fever, convulsions, bleeding and jaundice among others) was scored out of 14 items and presented as a percentage and graded as poor, average and good. Factors that contribute to the knowledge and practice in new–born care were analyzed using Pearson's chi squares.

## Results

Three hundred and seventy (370) care takers were studied and majority of these were the birth mothers of the children at 86%. The mean age of the of the caretakers was 26 SD 6 years and majority of the participant were in the age range of 21–35 years at 75%. The predominant family structure was nuclear at 77.6%. Only 6.5% of the mothers were living single and not in a stable relationship. Majority of the care givers had attained only primary school education at 38.1% while 7.6% had not received any formal education. About 73% of the caregivers were informally employed and 15% unemployed. Farming contributed 42% of the informal employment and 30.8% of the total employment of the respondents. Monthly income was mostly low with 74.6% earning below the poverty line at less than 210,000/ = Uganda shillings. Maternal parity was predominantly more than 4 children 37.6%) closely followed by the prime gravida at 35.7%. Mothers had a high antenatal care attendance of 97.6% and 89% had a birth companion during the birth process. Cesarean section mode of delivery was high at 55.7% and 96.2% of the deliveries were at a health facility Table 1.

### Essential new born care

Participant had variant practices in regard to essential newborn care. Majority (84.6%) of the respondents reported to not have put anything in the babies' eyes at birth. Among those who reported putting something in the eyes, 70% put the recommended tetracycline eye ointment, however, other things like breastmilk, water and saliva were reportedly put in the babies' eyes

**Table 1. Social demographic characteristics of the participants.**

| Variable | Frequency | Percentage |
|---|---|---|
| **Marital status** | | |
| Single | 24 | 6.49 |
| Married (in a relationship) | 346 | 93.51 |
| **Care givers' age (years)** | | |
| </ = 20 | 65 | 17.57 |
| >20–</ = 35 | 274 | 74.05 |
| >35 | 31 | 8.38 |
| **Education level** | | |
| None | 28 | 7.57 |
| Primary | 141 | 38.1 |
| High school | 137 | 37.03 |
| Tertiary | 64 | 17.30 |
| **Occupation** | | |
| Formal | 44 | 11.89 |
| Informal | 270 | 72.97 |
| None | 56 | 15.14 |
| **Income (Uganda Shillings)** | | |
| <210,000 | 276 | 74.59 |
| >210,000–500,000 | 73 | 19.73 |
| > 500,000 | 21 | 5.68 |
| **Mother's Parity** | | |
| Prime Gravida | 132 | 35.68 |
| 2–4 | 99 | 26.76 |
| >4 | 139 | 37.57 |
| **ANC attendance** | | |
| Yes | 361 | 97.57 |
| No | 9 | 2.43 |
| **Birth companion** | | |
| Yes | 330 | 89.19 |
| No | 40 | 10.81 |
| **Mode of delivery** | | |
| Cesarean section | 206 | 55.68 |
| Spontaneous Vaginal delivery | 164 | 44.32 |
| **Place of delivery** | | |
| Health facility | 356 | 96.22 |
| Home delivery | 14 | 3.78 |
| **Family Structure** | | |
| Extended | 82 | 22.40 |
| Nuclear | 284 | 77.60 |

at birth. In line with keeping hygiene, 76% of the mothers washed hands with soap and water before handling their babies, while 16.2% did not wash their hands at all. About 7.4% of the newborns received a bath within 24 hours of delivery. Use of herbs to bathe babies was reported by 19% of the care givers. Caregivers practiced adequate wrapping for the babies in 87% of the children. Majority of the caregivers regularly checked the baby's umbilical cord after birth, however 63.78% practiced inappropriate cord care practices. Among the care takers

who checked the cord, 36.5% report to have put something on the baby's cord and saliva was what was frequently used followed by baby oil and herbs at 14.6% and 13.9% respectively. Only 21% of the respondents reported initiation of breast feeding within 1 hour after birth and 15.4% after 24 hours of birth. Prelacteal feeds were given by 37.6% of the care givers, water being the commonest prelacteal feed followed by cow's milk at 40.4% and 18.4% respectively. Other prelacteal feeds included formula milk, glucose and soups. Most of the children are breastfed on demand (63.4%) while the others followed a timetabled format ranging from every 4 to 6 hours. 61.6% of the babies were reported to be vaccinated Table 2.

About 79% of the respondents reported giving treatment for the management of colic among babies where herbs were the commonest remedy given (47%) followed by medicines prescribed or bought at a pharmacy (27.3%).

There was reduced knowledge in identification of danger signs in the newborn by the caregivers (Fig 1). Fever was the commonest identified danger sign followed by inability to breast feed in a newborn baby.

Majority of the respondents had below average knowledge about danger signs in the newborn where 63% scored below 60% (Fig 2). Mean score for knowledge about danger sign was 44% SD 21%. Despite the low score, most respondents believed they would take a sick baby to health worker for help at 86%.

### Factors associated with newborn danger sign knowledge among caregivers

Caretaker's age and relationship of the caretaker with the child were found to have an association with knowledge of danger signs in the newborn baby at a P value of 0.019 and 0.009 respectively in a bivariate analysis Table 3.

### Discussion

This study set out to evaluate newborn care knowledge and practices and the associated factors among care givers of new-born babies at a regional referral hospital in South western Uganda and we found that participants had variant levels of knowledge in the various aspects of essential new born practices and low level of knowledge of the danger signs in a new born.

Health facility deliveries were high in this study at 96.2% which is similar to what was found in Southwestern Uganda, with health facility deliveries up to 90% in the same region [17]. In Tanzania and Angola, skilled birth attendance is lower at 64% and 50.7% respectively [18,19]. Antenatal care attendance was high and similar to a study done in Angola where ANC attendance was 96.8% [18]. There is a positive association of health facility delivery with antenatal care attendance [20]. Furthermore, our study was a hospital study and patients were more likely to come back to a health center if they had delivered from one for different health services like immunization and health care for their children.

WHO recommends essential newborn care including early initiation of breast feeding, keeping babies warm, recognition of neonatal danger signs and cord care, among others, as crucial in new born survival [11]. Contrary to WHO recommendations to initiate breastfeeding within an hour of birth, only 21.1% initiated breastfeeding within 1 hour after birth, and 15.45% of the newborns initiated breastfeeding after 24 hours in this study. This is much lower than what was found in a study done in India [21]. This disparity is explained by the fact that India has well established breast feeding programs which are absent in the current study setting. Furthermore, prelacteal feeds were administered in 37.6% of the newborn children. Similar findings were noted in a study done in Eastern Uganda where 35.6% of women practiced prelacteal feeding [22] This is lower than 64.7% that was found in Pakistan [23] probably because most of the birth in our study occurred in a health facility and caretakers may not

**Table 2. Essential newborn care practices.**

| Variable | Frequency | Percentage |
|---|---|---|
| **Breastfeeding** | | |
| **Initiation of breastfeeding** | | |
| >24hours | 57 | 15.45 |
| >4-24hours | 89 | 24.1 |
| >1-4hours | 145 | 39.3 |
| <1hour | 78 | 21.1 |
| **Prelacteal feeds** | | |
| No | 231 | 62.4 |
| Yes | 137 | 37.6 |
| **Thermal care** | | |
| Adequate | 325 | 87.8 |
| Inadequate | 45 | 12.2 |
| **Umbilical cord care** | | |
| **Checking Cord** | | |
| Don't check | 77 | 20.8 |
| Checks | 293 | 79.1 |
| **Cord care practices** | | |
| Right cord care practice \| | 134 | 36.2 |
| Wrong cord care practice | 236 | 63.8 |
| **What is put on the cord** | | |
| Ash | 16 | 11.7 |
| Baby oil | 20 | 14.6 |
| Herbs | 19 | 13.9 |
| Saliva | 56 | 40.9 |
| Powder | 12 | 8.8 |
| Others | 14 | 10.2 |
| **Hygiene (Hand washing)** | | |
| Don't wash | 60 | 16.2 |
| With water only | 29 | 7.8 |
| With water and Soap | 281 | 76 |
| **Eye care at birth** | | |
| No | 313 | 84.6 |
| Yes | 57 | 14.4 |
| **Bathing baby (Baby's first birth)** | | |
| Less than 24hours | 16 | 4.3 |
| More than 24 hours | 354 | 95.7 |
| **Colic management** | | |
| Nothing given | 78 | 21 |
| Herbs | 174 | 47.0 |
| Medicine | 101 | 27.3 |
| Others | 17 | 4.59 |
| **Immunization status of the babies** | | |
| Immunized | 228 | 61.6 |
| Not Immunized Yet | 142 | 38.4 |

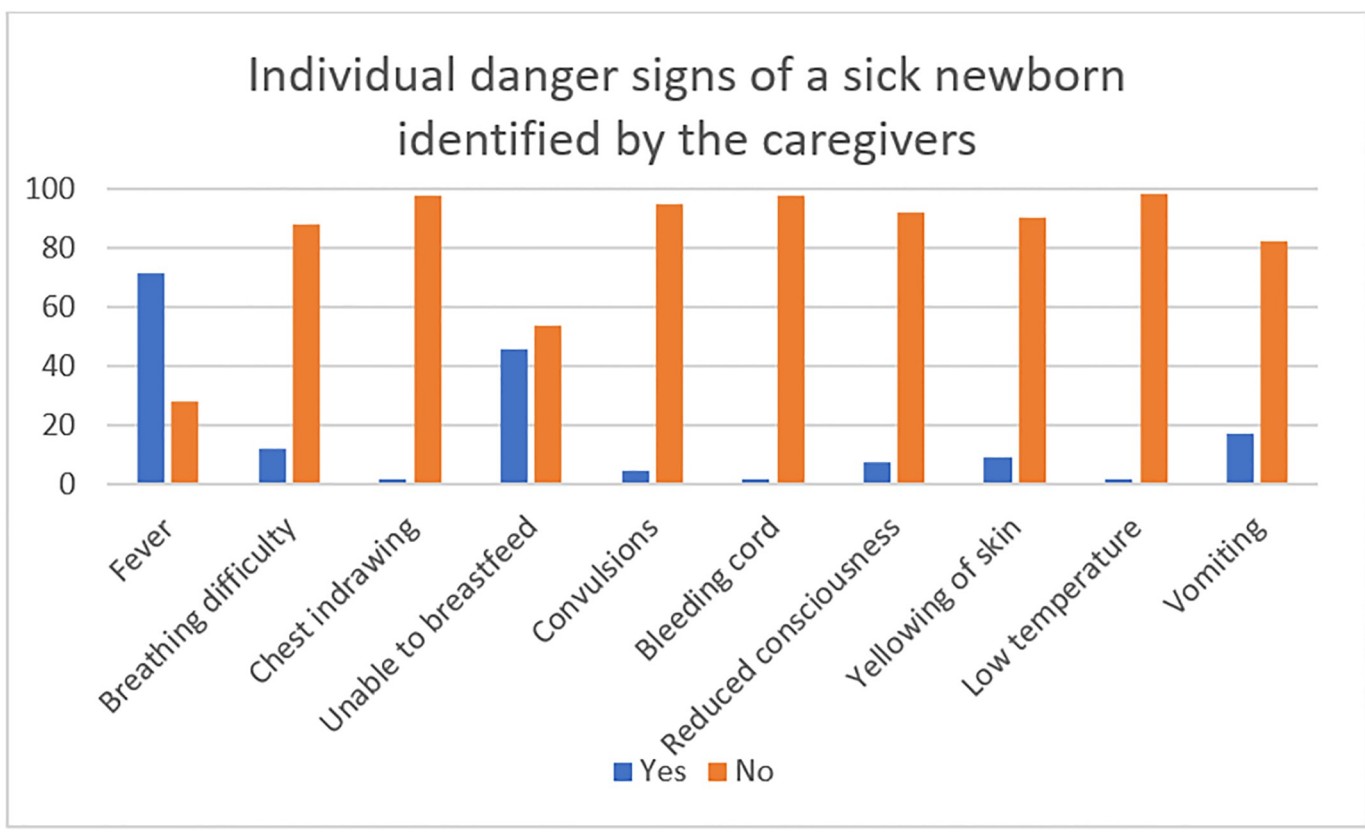

**Fig 1. Individual danger sign score of the caregivers.**

have had access to prelacteal feeds. However, the proportion of prelacteal feeds was higher than what was found in a multicenter study in East Africa [24] that included children that had been born 5 years earlier. The disparity could have been brought about by recall bias among caregivers compared to our study which included caretakers of infants from birth to six weeks of age.

Majority of the respondents purposefully checked their baby's cord, however reported wrong cord care practices with majority (63.8%) applying substances such as Ash, saliva, herbs and baby oil, which practice is contrary to standard and recommended care and puts newborn babies at risk of dying [25]. This however is similar to what has been found in Ghana (64.3%) [26] and Ethiopia with malpractice in cord care up to 66.9% [16].

In this study, care givers practiced adequate thermal care (87.8%) practicing wrapping the baby in multiple layer to keep the baby warm. This is higher than 60.9% found in a community study done in India [27] and 67.2% in a community study among adolescent mothers in Uganda [28]. This being a hospital based study, the respondents could have had health education that promoted adequate thermal care, but also it could be to the difference in cultural beliefs and that the Ugandan study included adolescent mothers who were not experienced in care on newborns. However, 24% of the newborns were reportedly given a bath within 24 hours of delivery. This finding is lower than what was found in a study done in Ethiopia where 32.5% of mothers had practiced early bathing of the newborn [29]. In Bangladesh, only 10.2% of mothers bathed their babies within 24 hours of delivery. These lower values may be attributed to differences in culture and beliefs surrounding birth in Africa and Asia.

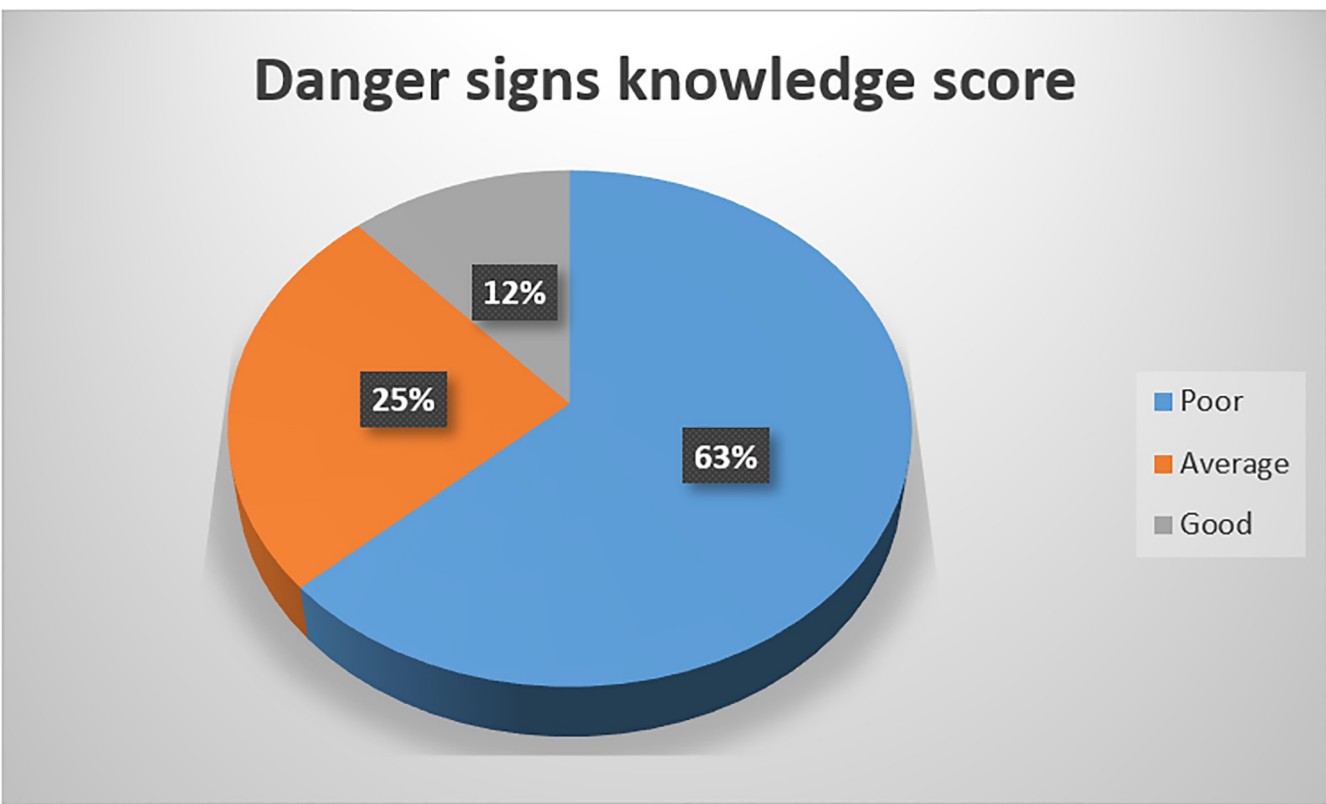

**Fig 2. Overall danger sign knowledge score among participants.**

About 38.4% of the newborn children were not immunized up to date despite the mean age being 7 days. This may be corresponding with the level of immunization coverage in the region [30], furthermore, since some of the newborn children were in hospital, they may have differed immunization until after discharge. In addition, the facility has one point of immunization where the infants have to be taken other than finding patients on maternity or paediatrics wards where they are born or admitted respectively. A study done in India found similarly low vaccination uptake of 47.4% for children less than 42 days [31] which may be attributed to access and attitudes of the caregivers to vaccination services.

Proper hand washing with soap and water before handling the baby was not practiced in 24% of the respondents. This may be due to cultural beliefs and lack of knowledge in regards to hygiene, but also lack of access to proper hand washing facilities at their homes and the newborn care facilities as identified in Nigeria and parts of Asia [32,33]. In addition, majority of the participants in the study were below poverty line which may pose a challenge in accessing basic needs like soap and clean water.

A big number of respondents were ignorant of eye care for the newborns with 84.6% unaware of the need to put any medication in the babies' eyes at birth. This is similar to studies done in India [34] which shares similar social and cultural beliefs to Uganda.

Regarding colic and its management, majority of the respondents (79%) had wrong practices that involved administration of herbs and over the counter medicines to alleviate the pain. This may be explained by lack of knowledge about appropriate measure and the caregivers' sense of helplessness when the child is in pain [35].

**Table 3. Factors associated with newborn danger sign knowledge among caregivers.**

| Variable | Poor Knowledge | Average Knowledge | Good Knowledge | P value |
|---|---|---|---|---|
| **Antenatal care attendance** | | | | 0.241 |
| <4 times | 124 (52.99) | 44 (47.31) | 27 (62.79) | |
| > 4 times | 110 (47.01) | 49 (52.69) | 16 (37.21) | |
| **Family Structure** | | | | 0.814 |
| Extended | 53 (23.04) | 21 (22.58) | 8 (18.6) | |
| Nuclear | 177 (76.96) | 72 (77.4) | 35(81.4) | |
| **Occupation** | | | | 0.192 |
| Formal | 23 (9.83) | 12 (12.9) | 9 (20.93) | |
| Informal | 173 (73.93) | 66 (70.79) | 31 (72.09) | |
| None | 38 (16.24) | 15(16.13) | 3 (6.98) | |
| **Income (Uganda Shillings)** | | | | 0.517 |
| <210,000 | 177 (75.64) | 70 (75.27) | 29 (67.44) | |
| >210,000 | 57 (24.36) | 23 (24.73) | 14 (32.56) | |
| **Mother's Parity** | | | | 0.865 |
| Prime Gravida | 84 (35.9) | 31(33.33) | 17 (39.53) | |
| 2–4 | 123 (52.56) | 50 (53.76) | 23 (53.49) | |
| >4 | 27 (11.4) | 12 (12.90) | 17 (39.53) | |
| **Marital Status** | | | | 0.816 |
| Married | 219 (93.59) | 86 (92.47) | 41 (95.35) | |
| Single | 15 (5.41) | 7 (7.53) | 2 (4.65) | |
| **Mode of delivery** | | | | 0.351 |
| Cesarean section | 128 (54.7) | 57 (61.29) | 21 (48.84) | |
| Spontaneous Vaginal delivery | 106 (45.3) | 36 (38.71) | 22 (51.16) | |
| **Place of delivery** | | | | 0.611 |
| Health facility | 226(96.58) | 88 (94.62) | 42 (97.67) | |
| Home delivery | 8 (3.42) | 5 (5.38) | 1 (2.33) | |
| **Family Structure** | | | | 0.814 |
| Extended | 82 | 22.40 | | |
| Nuclear | 284 | 77.60 | | |
| **Caregiver's age** | | | | 0.019 |
| <20 Years | 47 (20.09) | 14 (15.05) | 4(9.30) | |
| 21-30Years | 149 (63.68) | 63 (67.74) | 36(83.7) | |
| 31–40 Years | 37 (15.81) | 14 (15.05) | 1(2.33) | |
| >40 Years | 1 (0.43) | 2 (2.15) | 2(4.65) | |
| **Baby's Age** | | | | 0.089 |
| <7 days | 180 (76.92) | 82 (88.17) | 32 (74.42) | |
| 7–28 days | 43 (18.35) | 6 (6.45) | 8 (18.60) | |
| >28 days | 11 (4.70) | 5 (5.38) | 3 (6.98) | |
| **Relationship with child** | | | | 0.009 |
| Father | 5 (2.14) | 3 (3.23) | 2(4.65) | |
| Mother | 16 (6.84) | 16 (17.20) | 9 (20.93) | |
| Other | 213 (91.03) | 74 (79.57) | 32 (74.42) | |

Care giver's knowledge about newborn danger signs was low with a mean score of 44% which is similar to what has been found in Nigeria, Saudi Arabia and Ethiopia with majority of caregivers having low knowledge of danger signs in newborn children [36–38]. The low knowledge may be due to lack of targeted education programs for newborn caregivers focusing

on the sick newborn. The commonly recognized danger sign was fever at 71% of the respondents. Despite the relatively low knowledge about danger signs of a sick new born, majority of the respondents reported that they would seek care from a health worker and only 13 .1% seeking care and attention from other alternatives like relatives, spiritual leaders and herbalists in the event of a newborn.

In conclusion, this study reveals that there is various malpractice in the essential newborn care and low level of knowledge of danger sign in a new born among caregivers of newborn babies. There is an increased health center deliveries and antenatal care attendance among the respondents. This improved health services utilization could be used as an avenue to increase awareness about the malpractices in essential newborn care and also increase awareness about the danger signs in a newborn among caregivers of newborn children.

The strength of this study is that it highlighted the gaps in practice of essential newborn care by caregivers and reduced awareness among both parents and caregivers of newborn babies. This information could be used as baseline data in planning knowledge and practice enhancement programs for mothers and caregivers of new born babies.

The limitation of this study was that it was done in an institution and may not represent the caregivers and parents in the community. A community study could help evaluate any differences in knowledge and practice in care of the newborn.

## Author Contributions

**Conceptualization:** Dorah Nampijja, Stella Kyoyagala, Elias Kumbakumba.

**Data curation:** Dorah Nampijja, Stella Kyoyagala.

**Formal analysis:** Dorah Nampijja, Elias Kumbakumba.

**Funding acquisition:** Dorah Nampijja.

**Methodology:** Dorah Nampijja, Stella Kyoyagala, Elizabeth Najjingo, Josephine N. Najjuma, Onesmus Byamukama, Elias Kumbakumba.

**Project administration:** Dorah Nampijja, Stella Kyoyagala.

**Supervision:** Dorah Nampijja, Stella Kyoyagala, Elizabeth Najjingo, Josephine N. Najjuma, Jerome Kabakyenga, Elias Kumbakumba.

**Validation:** Dorah Nampijja, Stella Kyoyagala, Elizabeth Najjingo.

**Writing – original draft:** Dorah Nampijja, Stella Kyoyagala, Elizabeth Najjingo, Josephine N. Najjuma, Lydia Kyasimire, Elias Kumbakumba.

**Writing – review & editing:** Dorah Nampijja, Stella Kyoyagala, Elizabeth Najjingo, Josephine N. Najjuma, Onesmus Byamukama, Lydia Kyasimire.

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
