## [Decision Letter · Decision Letter 0]

20 Dec 2023

PONE-D-23-25688Newborn care knowledge and practices among care givers of new-born babies attending a regional referral hospital in Southwestern UgandaPLOS ONE

Dear Dr. Nampijja,

Thank you for submitting your manuscript to PLOS ONE. After careful consideration, we feel that it has merit but does not fully meet PLOS ONE’s publication criteria as it currently stands especially for Publication criteria Number 3. Therefore, we invite you to submit a revised version of the manuscript that addresses each of the points raised during the review process. The measurement and scoring of variables need to be better defined. Improve your discussion section, compare with studies in similar settings, discuss the implications of your findings. The manuscript will benefit from English Language editing services.

We look forward to receiving your revised manuscript.

Kind regards,

Adaoha Pearl Agu, MBBS, MSc, FMCPH

Academic Editor

PLOS ONE

Journal Requirements:

"I have read the journal's policy and the authors of this manuscript have no competing interests"

4. In this instance it seems there may be acceptable restrictions in place that prevent the public sharing of your minimal data. However, in line with our goal of ensuring long-term data availability to all interested researchers, PLOS’ Data Policy states that authors cannot be the sole named individuals responsible for ensuring data access (http://journals.plos.org/plosone/s/data-availability#loc-acceptable-data-sharing-methods).

Reviewers' comments:

Reviewer's Responses to Questions

**Comments to the Author**

1. Is the manuscript technically sound, and do the data support the conclusions?

Reviewer #1: Partly

Reviewer #2: Yes

2. Has the statistical analysis been performed appropriately and rigorously? 

Reviewer #1: No

Reviewer #2: Yes

3. Have the authors made all data underlying the findings in their manuscript fully available?

Reviewer #1: No

Reviewer #2: Yes

4. Is the manuscript presented in an intelligible fashion and written in standard English?

Reviewer #1: No

Reviewer #2: No

5. Review Comments to the Author

Reviewer #1: Global and Uganda data in background section is outdated. Can be updated consulting the Healthy Newborn Network platform (https://www.healthynewbornnetwork.org/).

The score for knowledge of danger signs is not described. In results, is the statistically significant correlation between knowledge of danger signs and caretaker's age and relationship to the baby, negative or positive? It is not clear.

There is too much information on the characteristics of the study site that is not relevant to the manuscript.

Some of the caretaker's inadequate practices require detailed descriptions (i.e. 'child scarification" and "false teeth extraction"), such as what is done and by whom. Also, it is not clear if the eye prophylaxis with antibiotic ointment refers in all cases to be done by health providers or by caretakers.

Since babies up to 6 weeks of age were included in the study, the description needs to be mentioned as "newborns and young infants).

The discussion section requires more work. There is no clarity on the purpose of comparing the findings with those in many other countries. There is no need to repeat the results, unless there is an analysis of possible causes and proposed potential solutions.

Reviewer #2: Minor

Inconsistent spelling of newborn / new born / new-born

Overall

Congratulations on this important paper addressing a very essential area for improvement about low levels of knowledge of danger signs, both when to seek care, but also focus on how the newborn is handled by the caregiver.

I believe the information is important, but the paper needs major revision. Overall, there is quite a few language issues and/or grammatical errors that could be improved. I will not comment on those, just the information in the paper.

Introduction

Recommend using updated numbers for neonatal mortality of 2.3 million newborns (Lawn et. Al

https://www.ncbi.nlm.nih.gov/pmc/articles/PMC10614465/

Ending Preventable Neonatal Deaths: Multicountry Evidence to Inform Accelerated Progress to the Sustainable Development Goal by 2030

The Sustainable Development Goal (SDG) 3.2 aims for every country to reach a neonatal mortality rate (NMR) of ≤12/1,000 live births by 2030. More than 60 countries are off track, and 2.3 million newborns still die each year. Urgent action is needed, ...

www.ncbi.nlm.nih.gov

Recommends calling it the neonatal period – hence 28 days of life, and not one month of age.

Sentence “This knowledge is often wrong and inappropriate” - would suggest more neutral less judgemental wording,

Do you have any information on the information given to the parents on discharge, if any about danger signs, ENC and when to seeks care? Could be included here or in the population section in the methods section.

Methods

The inclusion criteria could be specified more, also a flow chart of how many of the caregivers were included from immunization clinics, maternity ward and NICU.

Results

Nuclear family structure might be difficult to understand for readers from other cultures, would suggest monogamy or something more explanatory.

Is information available from how many of the caregivers who had a sick newborn who was admitted. Not clear from the text or table?

Percentages for most ENC are reported, but not for umbilical cord care. Would recommend including the number in the text.

Discussion

Oveall some good points.

Regarding discussion on prelacteal feeding, I think it should be elaborated with references to other more similar studies from similar settings.

In the paragraph on wrapping, the comparison is not well written. It is a lower percentage that is not wrapped properly compared to India, the sentence isa bit confusing.

Regarding knowledge of danger signs the discussion compares to Ethiopia, which is good, but Saudi Arabia has a very different population, and health system, so might not be the most relevant comparison. Would recommend exploring literature from other more similar countries.

Of those who seek care, it is not clear if it is someone who did seek care or who would in case of a danger sign. Did 71% of the newborns have fever? That seems like a very high number when the study population is a mixture of parents found in immunization clinics, maternity ward and newborn unit

6. PLOS authors have the option to publish the peer review history of their article (what does this mean?). If published, this will include your full peer review and any attached files.

Reviewer #1: **Yes: **Goldy Mazia, MD MPH. Senior Newborn Health Advisor. Save the Children US

Reviewer #2: **Yes: **Dr. Charlotte Carina Holm-Hansen

---

## [Author Response · Author response to Decision Letter 0]

19 Feb 2024

I have worked on all the comments that were given to me by the reviewer and re attached the revised manuscript alongside a rebuttal letter for your consideration

---

## [Editor Report · Decision Letter 1]

20 Mar 2024

Newborn care knowledge and practices among care givers of new-born babies attending a regional referral hospital in Southwestern Uganda

PONE-D-23-25688R1

Dear Dr. Nampijja,

We’re pleased to inform you that your manuscript has been judged scientifically suitable for publication and will be formally accepted for publication once it meets all outstanding technical requirements.

An invoice for payment will follow shortly after the formal acceptance. To ensure an efficient process, please log into Editorial Manager at Editorial Manager® , click the 'Update My Information' link at the top of the page, and double check that your user information is up-to-date. If you have any billing related questions, please contact our Author Billing department directly at authorbilling@plos.org.

Kind regards,

Adaoha Pearl Agu, MBBS, MSc, FMCPH

Academic Editor

PLOS ONE
---

## [Editor Report · Acceptance letter]

26 Apr 2024

PONE-D-23-25688R1 

PLOS ONE

Dear Dr. Nampijja, 

I'm pleased to inform you that your manuscript has been deemed suitable for publication in PLOS ONE. Congratulations! Your manuscript is now being handed over to our production team.

Kind regards, 

on behalf of

Dr. Adaoha Pearl Agu 

Academic Editor

PLOS ONE